# Suppression of TRPV1/TRPM8/P2Y Nociceptors by Withametelin via Downregulating MAPK Signaling in Mouse Model of Vincristine-Induced Neuropathic Pain

**DOI:** 10.3390/ijms22116084

**Published:** 2021-06-04

**Authors:** Adnan Khan, Bushra Shal, Ashraf Ullah Khan, Rahim Ullah, Muhammad Waleed Baig, Ihsan ul Haq, Eun Kyoung Seo, Salman Khan

**Affiliations:** 1Pharmacological Sciences Research Lab, Department of Pharmacy, Faculty of Biological Sciences, Quaid-i-Azam University, Islamabad 45320, Pakistan; adkhan165sbbu@gmail.com (A.K.); bushra.shal@gmail.com (B.S.); ashrafwazir6@gmail.com (A.U.K.); mwbg7@yahoo.com (M.W.B.); ihsn99@yahoo.com (I.u.H.); 2Department of Pharmacy, University of Peshawar, Peshawar 25120, Pakistan; Rphrahimullah@gmail.com; 3Graduate School of Pharmaceutical Sciences, College of Pharmacy, Ewha Womans University, Seoul 03760, Korea

**Keywords:** neuropathic pain, vincristine, withametelin, TRPV1/TRPM8, MAPK, Bax/Bcl-2

## Abstract

Vincristine (VCR) is a widely used chemotherapy drug that induced peripheral painful neuropathy. Yet, it still lacks an ideal therapeutic strategy. The transient receptor potential (TRP) channels, purinergic receptor (P2Y), and mitogen-activated protein kinase (MAPK) signaling play a crucial role in the pathogenesis of neuropathic pain. Withametelin (WMT), a potential Phytosteroid isolated from *datura innoxa*, exhibits remarkable neuroprotective properties. The present investigation was designed to explore the effect of withametelin on VCR-induced neuropathic pain and its underlying molecular mechanism. Initially, the neuroprotective potential of WMT was confirmed against hydrogen peroxide (H_2_O_2_)-induced PC12 cells. To develop potential candidates for neuropathic pain treatment, a VCR-induced neuropathic pain model was established. Vincristine (75 μg/kg) was administered intraperitoneally (i.p.) for 10 consecutive days (day 1–10) for the induction of neuropathic pain. Gabapentin (GBP) (60 mg/kg, i.p.) and withametelin (0.1 and 1 mg/kg i.p.) treatments were given after the completion of VCR injection on the 11th day up to 21 days. The results revealed that WMT significantly reduced VCR-induced pain hypersensitivity, including mechanical allodynia, cold allodynia, and thermal hyperalgesia. It reversed the VCR-induced histopathological changes in the brain, spinal cord, and sciatic nerve. It inhibited VCR-induced changes in the biochemical composition of the myelin sheath of the sciatic nerve. It markedly downregulated the expression levels of TRPV1 (transient receptor potential vanilloid 1); TRPM8 (Transient receptor potential melastatin 8); and P2Y nociceptors and MAPKs signaling, including ERK (Extracellular Signal-Regulated Kinase), JNK (c-Jun *N*-terminal kinase), and p-38 in the spinal cord. It suppressed apoptosis by regulating Bax (Bcl2-associated X-protein), Bcl-2 (B-cell-lymphoma-2), and Caspase-3 expression. It considerably attenuated inflammatory cytokines, oxidative stress, and genotoxicity. This study suggests that WMT treatment suppressed vincristine-induced neuropathic pain by targeting the TRPV1/TRPM8/P2Y nociceptors and MAPK signaling.

## 1. Introduction

Neuropathic pain, also referred to as nerve pain, results from different types of nerve injuries, such as chemotherapy-induced neuropathy, traumatic nerve injury, post-herpetic neuralgia, and diabetic neuropathy [1,2]. Chemotherapy-induced peripheral neuropathy (CIPN) is a typical side effect produced during antitumor treatments [3]. Vincristine (VCR) is an effective chemotherapeutic drug indicated in the treatment of many types of cancer, such as lymphomas, leukemia, breast cancer, and brain tumors [4]. However, VCR-induced neuropathy is a major dose-limiting side effect and causes the premature termination of cancer therapy and, thus, results in a great impact on the survival of cancer patients [5]. A hallmark of VCR-induced neuropathy is described by the degeneration of distal sensory axons, demyelination, and loss of nerve fibers [6]. Presently, the therapeutic strategies used against VCR-induced neuropathic pain are limited to anticonvulsants, opioids, and tricyclic antidepressants. However, these treatments are associated with a wide spectrum of adverse effects that limit their satisfactory clinical use in the amelioration of peripheral neuropathy [2]. Therefore, novel drugs or therapies that suppress CIPN caused by vincristine without attenuating its anticancer effects are urgently required.

For the last two decades, research has been carried out to explore the underlying mechanisms of VCR-induced neuropathic pain in rodents model. It is well-recognized that TRP nociceptors—in particular, TRPV1 and TRPM8—play a crucial role in chemotherapeutics-induced mechanical and thermal hypersensitivity in rodents [7,8]. To date, the TRPV1 channel has become the most thoroughly studied TRP channel and has been triggered by a variety of stimuli, such as temperature (43–52 °C), pH, and a variety of endogenous and exogenous compounds [9]. TRPV1 is found in several areas of the CNS, most notably in the dorsal horn of the spinal cord, where it regulates the synaptic transmission of nociceptive signals from the periphery [10]. It is well-accepted that TRPV1 plays a significant role in the hypersensitivity to thermal, chemical, and mechanical stimuli, which has been linked to peripheral neuronal damage [11]. Numerous studies have reported that the pharmacological inhibition of TRPV1 relieves pain hypersensitivity in rodent models of neuropathic pain [12]. Similarly, the TRPM8 ion channel is another important TRP family member that responds to mild and noxious cold (<25 °C) temperatures [13]. Several lines of evidence have indicated that the expression of TRPM8 increases in the spinal cord and dorsal root ganglion (DRG) in animal models of neuropathic pain and is thought of as one of the potential targets for neuropathic pain treatment [7]. Numerous studies have found an association between the degree of P2Y receptor expression and the generation of neuropathic pain. It is well-established that the pharmacological inhibition of the P2Y receptor relieves pain hypersensitivity in animal models of neuropathic pain [14].

The mitogen-activated protein kinases (MAPKs), including the p38 mitogen-activated protein kinase (p38), c-Jun *N*-terminal kinase (JNK), and extracellular-regulated kinase (ERK) signaling, play a crucial role in the pathophysiology of VCR-induced neuropathy [15]. Several lines of evidence have indicated that ERK and p38-MAPK are stimulated and their expression levels are increased in the spinal dorsal horn after a peripheral nerve injury [16,17,18]. The inhibitors of MAPKs have been known to ameliorate neuropathic pain [19]. It is well-known that MAPK plays a crucial role in neuropathic pain via the transcriptional regulation of P2Y receptors [20]. Correspondingly, TRPV1 expression is also transcriptionally regulated by the activation of MAPK signaling [21]. Several lines of evidence have indicated that VCR induces an inflammatory cascade involving p38-MAPK signaling that stimulates the downstream signaling of inflammatory mediators, including tumor necrosis factor-alpha (TNF-α), interleukin-1 beta (IL-1β), and nitric oxide [22,23]. These inflammatory mediators directly induce pain hypersensitivity in the nerves, along with the stimulation of the gene expression of inflammatory factors [22,24]. Therefore, effective treatments to suppress inflammatory cascades might play an important role in treating neuropathic pain.

Growing evidence has indicated that JNK signaling plays a critical role in apoptosis. JNK inactivates antiapoptotic Bcl-2 proteins and activates caspase-3/Bax (apoptotic proteins), which contributes to apoptosis [25,26]. It is well-recognized that the upregulation of caspase-3/Bax and downregulation of Bcl-2 (an antiapoptotic protein) substantiates the activation of the apoptotic pathways and, thus, accelerates the death process following VCR-induced peripheral nerve damage [27]. Furthermore, the involvement of oxidative stress is widely reported in the pathogenesis of VCR-induced neuropathy and is thought to be one of the major contributors to neuropathic pain [1]. It is well-established that VCR-induced neuropathic pain is described by an elevated level of oxidative stress markers, such as malondialdehyde (MDA), myeloperoxidase (MPO), and nitric oxide (NO), and the depletion of antioxidants, including glutathione (GSH), catalase (CAT), glutathione-S-transferases (GST), and superoxide dismutase (SOD) in the spinal cord and sciatic nerve of mice [2,28].

Withanolides are steroidal lactones that possess significant immunomodulatory, antimicrobial, antioxidant, and anti-inflammatory properties [29]. Withametelin (WMT) is a potent withanolide derivative obtained from the leaves of *Datura innoxa* that exhibit diverse pharmacological properties, including anti-inflammatory, antioxidant, antidepressant, and anticancer [30,31,32]. The leaves of *Datura innoxa* have been used in traditional medicine to treat seizures and nerve pain (facial pain and headache) [33]. The effect of withametelin on neuropathic pain has not been investigated yet. Therefore, the present study was designed to explore the effect of withametelin on VCR-induced neuropathic pain and establish the involvement of TRP/P2Y nociceptors, MAPK signaling, apoptosis, and oxidative stress using biochemical, histopathological, immunohistochemical, and molecular techniques.

## 2. Results

### 2.1. Effects of WMT on Cell Viability, GSH, and NO Level in H_2_O_2_-Induced PC12 Cells

An MTT (3-(4,5-Dimethylthiazol-2-yl)-2,5-Diphenyltetrazolium Bromide) assay was used to explore the neuroprotective effect of WMT against H_2_O_2_-induced cytotoxicity. PC12 cells were pretreated with WMT (1, 10, 50, and 100 µM) and curcumin (50 µM) concentrations for 2 h before treatment with 200 µM H_2_O_2_. WMT markedly (*p* < 0.001) attenuated H_2_O_2_-induced cell viability loss in a dose-dependent manner compared to H_2_O_2_ alone. Furthermore, the levels of NO and GSH in PC12 cells were measured to examine the effects of WMT on H_2_O_2_-induced oxidative stress. The results revealed that WMT significantly (*p* < 0.001) suppressed H_2_O_2_-induced oxidative stress by increasing the level of GSH but decreasing the NO level in a dose-dependent manner compared to H_2_O_2_ alone (Figure 1).

### 2.2. WMT Attenuated VCR-Induced Pain Hypersensitivity in Mice

Vincristine (75 μg/kg, i.p.) administration for 10 consecutive days resulted in a marked (*p* < 0.001) increase in pain hypersensitivity, as reflected by the decreased pain threshold in mechanical allodynia, cold allodynia, and thermal hyperalgesia. The WMT treatment significantly (*p* < 0.001) reversed VCR-induced mechanical allodynia, cold allodynia, and thermal hyperalgesia (Figure 2).

### 2.3. WMT Suppressed VCR-Induced Oxidative Stress in the Brain, Spinal Cord, and Sciatic Nerve

To explore the antioxidant potential of WMT against VCR-induced oxidative stress, the level of antioxidant (GSH, GST, Catalase, and SOD); MPO activity; and MDA level were determined in the brain regions (ACC, PAG, HC, IC, and amygdala); spinal cord (L4–L6); and sciatic nerve. VCR markedly (*p* < 0.001) reduced the antioxidants levels in the mice brain, spinal cord, and sciatic nerve. The WMT treatment considerably improved the antioxidants levels in the brain, spinal cord, and sciatic nerve. Correspondingly, VCR strikingly increased the levels of MDA and MPO in the mice brain, spinal cord, and sciatic nerve. The WMT treatment remarkably (*p* < 0.001) reduced the MDA and MPO levels in the mice brain, spinal cord, and sciatic nerve (Figure 3 and Figure 4).

### 2.4. WMT Ameliorated VCR-Induced Inflammation in the Brain, Spinal Cord, and Sciatic Nerve

To investigate the anti-neuroinflammatory potential of WMT against VCR-induced inflammation, the levels of the inflammatory cytokines (IL-1β and TNF-α) and NO production were measured in the mice brain regions (ACC, PAG, HC, IC, and amygdala); spinal cord (L4–L6); and sciatic nerve. VCR significantly increased (*p* < 0.001) the IL-1β and TNF-α levels in the mice brain, spinal cord, and sciatic nerve. The WMT treatment markedly (*p* < 0.001) suppressed the VCR-induced inflammatory cytokines level. Correspondingly, VCR strikingly increased NO production in the mice brain, spinal cord, and sciatic nerve. The WMT treatment markedly (*p* < 0.001) reduced the NO production in the mice brain, spinal cord, and sciatic nerve (Figure 5).

### 2.5. WMT Inhibited VCR-Induced Histopathological Changes in the Sciatic Nerve

To explore the VCR-induced histopathological changes in the sciatic nerve, H&E and trichome staining were performed. Vincristine causes substantial (*p* < 0.001) nerve degeneration and derangement, demyelination, and axonal swelling in comparison to the normal group. WMT markedly (*p* < 0.001) attenuated vincristine-induced histopathological changes, such as nerve degeneration and derangement, demyelination, and axonal swelling in the sciatic nerve (Figure 6).

### 2.6. WMT Reversed VCR-Induced Histopathological Changes in the Spinal Cord

To observe VCR-induced histopathological changes, transverse sections of the spinal cord tissue were subjected to H&E staining. A striking (*p* < 0.001) increase in the cellular infiltration/inflammatory lesion and vacuolar changes were observed in the transverse section of the spinal cord of the VCR group. As compared to the VCR group, WMT showed a marked (*p* < 0.001) reduction in the cellular infiltration/inflammatory lesion and vacuolar changes in the transverse section of the spinal cord (Figure 7). To further investigate VCR-induced histopathological alterations in the spinal cord, longitudinal sections of the spinal cord tissue were also subjected to H&E staining. A marked (*p* < 0.001) increase in cellular infiltration, inflammatory lesion, and vacuolation was observed in the VCR group. However, WMT showed a reduction in cellular infiltration, inflammatory lesion, and vacuolation in the longitudinal section of the spinal cord compared to the VCR group (Figure 7).

### 2.7. WMT Inhibited VCR-Induced Histopathological Changes in Brain

H&E staining was performed to investigate the VCR-induced histopathological changes in the brain. In comparison to the normal group, VCR significantly (*p* < 0.001) reduced the granular layer of the hippocampal dentate gyrus. WMT inhibited VCR-induced dentate gyrus thickness reduction. In the same way, the formation of inflammatory plaques in the cerebral cortex was observed in the VCR group. However, the WMT treatment markedly (*p* < 0.001) suppressed the formation of VCR-associated inflammatory plaques in the cerebral cortex (Figure 8).

### 2.8. FTIR Spectroscopic and DSC Analysis of the Sciatic Nerve

In the present research, we performed FTIR spectroscopy to characterize the lipid, protein, nucleic acid, and carbohydrate content of the myelin sheath of the sciatic nerve. The representative IR absorption spectra were obtained within the spectral range of 450–4000 cm^−1^. The results indicated that marked variations in the biomolecular composition, including protein oxidative damage, lipid peroxidation, increase in the nucleic acids/carbonyl content, and decrease in the lipid/protein contents were observed. WMT considerably lowered the VCR-induced changes in the biochemical composition of the sciatic nerve’s myelin sheath. The wavenumber of the band was shifted significantly to a higher value compared to the VCR group (Figure 9A). The assignments of the absorption peaks observed in the spectra are listed in Table 1.

DSC is a highly sensitive method for precisely recording the thermal denaturation transitions of a protein. To explore the protective potential of WMT, a DSC analysis of the sciatic nerve was performed. In DSC, thermal events of the sciatic nerve were recorded. A complex exothermic–endothermic denaturation transition was observed in the VCR group during the first cycle of the heat. The present results indicated that there was substantial protein damage in the sciatic nerve of the VCR group, as shown by a complex exothermic–endothermic denaturation transition. However, the treatment with WMT inhibited VCR-induced protein damage (Figure 9B).

### 2.9. WMT Ameliorated VCR-Induced Genotoxic Effect in the Sciatic Nerve

To investigate whether the WMT treatment reversed the genotoxic effect of a VCR injury in the sciatic nerve, a comet assay was performed. A significant (*p* < 0.001) increase in the tail length and % of DNA in the tail was observed in the VCR group, which indicates neuronal DNA damage in the sciatic nerve. The WMT treatment markedly (*p* < 0.001) alleviated the VCR-induced genotoxic effect by reducing the tail length and % of DNA in the tail (Figure 10).

### 2.10. WMT Reduced the mRNA Expression Level of TRPV1/TRPM8/P2Y Nociceptors and JNK in the Spinal Cord after VCR Administration

It is well-established that the TRP channels (TRPV1 and TRPM8), P2Y, and JNK activation play a vital role in the development and maintenance of chemotherapy-induced neuropathic pain. To investigate the possible role of TRPV1/TRPM8/P2Y and JNK in VCR-induced neuropathic pain, a qRT-PCR analysis was performed. Our results demonstrated that the mRNA expression levels of TRPV1/TRPM8/P2Y and JNK were significantly (*p* < 0.001) increased in the spinal cord of the VCR group. However, the WMT treatment strikingly (*p* < 0.001) reduced the mRNA expression levels of TRPV1/TRPM8/P2Y and JNK in the spinal cord of VCR-treated mice (Figure 11). The list of forward and reverse primers is shown in Table 2.

### 2.11. WMT downregulates TRPV1/TRPM8/P2Y Nociceptors in the Spinal Cord after VCR Administration

To determine the possible role of the TRP channels and purinergic receptors in VCR-induced neuropathic pain, the TRPV1/TRPM8/P2Y protein expressions were measured in the mice spinal cord. The results indicated that a vincristine injection induced a marked (*p* < 0.001) increase in TRPV1/TRPM8/P2Y immunoreactivity compared to the control group. However, WMT significantly (*p* < 0.001) downregulated the TRPV1/TRPM8/P2Y expression in the mice spinal cord as compared to the vincristine group (Figure 12).

### 2.12. WMT Attenuates the ERK/JNK/p38 Expression in the Spinal Cord after VCR Administration

Since MAPK signaling has been implicated in the pathogenesis of neuropathic pain, we investigated the expression level of the MAPK (ERK/JNK/p38) protein in mice spinal cords. In comparison to the control group, vincristine injection induced a substantial (*p* < 0.001) increase in ERK/JNK/p38 immunoreactivity in mice spinal cords. WMT significantly (*p* < 0.001) decreases ERK/JNK/p38 expression in the mice spinal cord as compared to the vincristine group (Figure 13).

### 2.13. WMT Suppresses Apoptosis by Regulating the Bax/Bcl-2/Caspase-3 Expression in the Spinal Cord after VCR Administration

To evaluate the antiapoptotic potential of WMT against VCR-induced apoptosis, we measured the expression of Bax, Bcl-2, and caspase-3 in the spinal cord. The results showed that the WMT treatment significantly (*p* < 0.001) decreased the expression of the proapoptotic protein (Bax/caspase-3) but increased the expression of the antiapoptotic protein (Bcl-2) in the spinal cord of VCR-treated mice (Figure 14).

### 2.14. Docking Interaction with WMT

The interaction of WMT with target proteins was investigated using molecular docking studies based on an IHC analysis. The results of the docking analysis were visualized using discovery studio_2019 and presented as 3D and 2D views. The results of the docking studies showed that WMT promisingly interacts with TRPV1, TRPM8, P2Y, p38, JNK, and caspase-3 via a hydrogen bond and hydrophobic interaction (Figure 15). The binding energies of the protein targets after docking with WMT were measured as −9.6 Kcal/mol for TRPV1, −8.4 Kcal/mol for TRPM8, −8.6 Kcal/mol for P2Y, −8.9 Kcal/mol for p38, −8.3 Kcal/mol for JNK, and −10.2 Kcal/mol for caspase-3, as shown in the heatmap (Figure 15). Table 3 shows the binding energies, hydrogen bonds, and hydrophobic interactions of WMT with various protein targets.

## 3. Discussion

The present investigation explores the neuroprotective potential of WMT against vincristine-induced neuropathic pain, along with the possible underlying molecular mechanism using biochemical, histopathological, immunohistochemical, and molecular techniques. Based on the findings of the current study, it was observed that WMT attenuated vincristine-induced pain hypersensitivity, suggesting its antinociceptive potential in painful neuropathy. More prominently, WMT showed neuroprotective potential through the modulation of *TRPV1*/TRPM8/P*2Y*, MAPK, and Bax/Bcl-2/caspase-3 signaling in the spinal cord.

Preliminary, the neuroprotective potential of WMT was confirmed against H_2_O_2_-induced PC12 cells. It was noticed that a pretreatment with WMT markedly inhibited the loss of cell viability and H_2_O_2_-induced oxidative stress in a dose-dependent manner compared to H_2_O_2_ alone. These findings indicated that WMT had a significant protective potential against H_2_O_2_-induced cytotoxicity in PC12 cells via the inhibition of oxidative stress. After the confirmation of WMT’s protective activity in an in vitro model, a well-known in vivo model of vincristine-induced peripheral neuropathy was established to explore WMT’s mechanisms as a possible candidate for the treatment of neuropathic pain. The results of the behavioral study showed that WMT ameliorated vincristine-induced pain hypersensitivity (i.e., cold allodynia, mechanical allodynia, and thermal hyperalgesia) in a dose-dependent manner. The behavioral alteration caused by vincristine in this study is consistent with previous studies [28,34].

The role of oxidative stress in the progression of chemotherapy-induced neuropathy has been well-reported, and it is thought to be one of the major factors of neuropathic pain [1,4]. In this study, it was noticed that VCR-induced oxidative stress by the depletion of antioxidants and increased the levels of MDA, MPO, and NO production in the brain, spinal cord, and sciatic nerve. However, the WMT treatment substantially increased antioxidants levels while suppressing the NO, MPO, and MDA levels in the brain, spinal cord, and sciatic nerve. These results were in line with previous studies [3,4]. Besides, proinflammatory cytokines, including tumor necrosis factor-alpha (TNF-α) and interleukin-1 beta (IL-1β), also play a crucial role in VCR-induced neuropathic pain [2]. The levels of VCR-induced inflammatory cytokines were significantly reduced by the WMT treatment. These findings were consistent with earlier reports [4,35,36].

The fact that peripheral neuropathic pain is linked to substantial histopathological changes in the brain, spinal cord, and the sciatic nerve is well-established [2,37,38]. H&E staining was used to investigate VCR-induced histopathological changes in the brain and spinal cord. H&E and trichome staining were used to examine the histopathology of the sciatic nerve. The results demonstrated that WMT markedly inhibited the vincristine-induced histopathological changes in the brain, spinal cord, and sciatic nerve. Next, a comet assay was used to explore the VCR-induced genotoxic effects in the sciatic nerve. The results indicated that WMT alleviated the VCR-induced genotoxic effects (DNA damage) in the sciatic nerve.

In the present study, Fourier-transform infrared (FTIR) spectroscopy was used to elucidate the lipid, protein, nucleic acid, and carbohydrate contents of the myelin sheath of the sciatic nerve in the VCR-induced model. The results of the FTIR analysis indicated that WMT exhibited a neuroprotective effect on the VCR-induced changes in the biochemical composition of the sciatic nerve’s myelin sheath, which shows its potential to treat nerve damage or neuropathy. The present results were consistent with previous studies that also demonstrated that damage to the myelin sheath via alterations in the biochemical composition of the sciatic nerve tissue contributes to peripheral neuropathy [37]. Differential scanning calorimetry (DSC) is a high-tech thermoanalytical technique for detecting and characterizing changes in the tissues at the molecular and supramolecular levels that are linked to drug-induced neurodegeneration [39,40]. The current results of the DSC analysis demonstrated that there were remarkable changes in the secondary structure of the protein of the sciatic nerve of the VCR group. However, the WMT treatment remarkably preserved the secondary structures of the proteins.

Several lines of evidence indicated that the TRP channels i.e., vanilloid 1 (TRPV1) and melastatin 8 (TRPM8) play a pivotal role in chemotherapy-induced neuropathic pain [41,42]. Correspondingly, recent neurobiological studies have indicated that the P2Y receptor is involved in the development and maintenance of neuropathic pain [20]. It has been reported that the activation of the P2Y receptors induces the release of inflammatory cytokines in the spinal cord following a nerve injury in neuropathic pain [43]. It is well-recognized that MAPK induces the expression of P2Y receptors, but the MAPK inhibitor significantly suppresses the P2Y receptors and attenuated the mechanical pain hypersensitivity. Thus, MAPK plays a crucial role in neuropathic pain via the transcriptional regulation of the P2Y receptor [20]. Numerous studies have reported that MAPK signaling plays a vital role in the regulation of TRPV1 nociceptors. The activation of p38-MAPK results in the upregulation of TRPV1 in the peripheral nerve terminals [21,44]. To determine the possible role of TRPV1/TRPM8/P2Y protein expression in the antinociceptive potential of WMT against-VCR-induced neuropathic pain, the mRNA expression levels of TRPV1/TRPM8/P2Y and JNK were measured. The results indicated that WMT markedly reduced the mRNA expression levels of TRPV1/TRPM8/P2Y and JNK in the spinal cord. These results suggest that the downregulation of the mRNA expression levels of the TRPV1/TRPM8/P2Y and JNK proteins by WMT in the spinal cord are likely to have an important role in the amelioration of VCR-induced painful neuropathy.

To further strengthen the participation of the spinal TRPV1/TRPM8/P2Y receptors in the development of VCR-induced neuropathic pain, we measured the protein expression of the TRPV1/TRPM8/P2Y receptors using immunohistochemistry. A significant increase in the expression of the TRPV1/TRPM8/P2Y proteins was observed in the spinal cord of the VCR group. However, TRPV1/TRPM8/P2Y protein expression in the mice spinal cords was substantially reduced by WMT. Our data strongly suggests that spinal TRPV1/TRPM8/P2Y nociceptors participate in the development of neuropathic pain. The current results were reliable with other studies that suggested that antagonists of TRPV1/TRPM8 and P2Y protect against chemotherapeutic-induced inflammation, oxidative stress, thermal hyperalgesia, cold allodynia, and mechanical allodynia [14,41,45].

It is well-established that MAPK signaling plays a crucial role in the pathogenesis of neuropathic pain. In mammalian cells, three MAPK signaling pathways, including ERK, JNK, and p38-MAPK, are reported to be involved in the development of pain hypersensitivity [46]. WMT markedly decreased ERK/JNK/p38 expression in the mice spinal cord as compared to the vincristine group. The present results were reliable with other studies, which suggested that the inhibitors of MAPKs contributed to the attenuation of neuropathic pain [15]. Mounting evidence has indicated that JNK signaling plays a pivotal role in apoptosis. JNK activates caspase-3/Bax (apoptotic protein) and inactivates the antiapoptotic Bcl-2 protein, resulting in apoptosis [25,26]. It is well-accepted that the upregulation of caspase-3/Bax and downregulation of Bcl-2 induces apoptosis and, subsequently, VCR-induced peripheral nerve damage [27]. In the present study, the WMT treatment reduced the expression of the proapoptotic proteins (Bax/caspase-3) but enhanced the expression of the antiapoptotic protein (Bcl-2) in the spinal cord of VCR-treated mice, thus reducing apoptotic cell death. These findings were consistent with earlier reports [4].

Molecular docking is the most useful technique to explore the possible affinity between a ligand and protein complex [39]. The above-mentioned results suggest that the WMT treatment suppressed vincristine-induced neuropathic pain by targeting the TRPV1/TRPM8/P2Y nociceptors and MAPK signaling. Therefore, in the current study, molecular docking was performed for WMT–TRPV1, WMT–TRPM8, WMT–P2Y, WMT–p38, WMT–JNK, and WMT–Caspase-3 to explore the possible affinity of withametelin with the mentioned protein targets. We identified the molecular interaction patterns of withametelin binding in each target protein. E-value, H-bonding, and hydrophobic interactions play a vital role in the assessment of the binding affinity of ligand and protein complexes. The molecular interactions were elaborated in terms of the hydrogen bonds and hydrophobic interactions. Hydrogen bonding and hydrophobic interactions provide valuable strength for drug–receptor complex stabilization. Hydrogen bond formation is crucial for recognition, stabilization, and molecular movements. Correspondingly, hydrophobic interactions are necessary for increasing ligands’ affinities towards protein receptors [47]. Our thorough analysis of the computational docking indicated that WMT binds each target protein by H-bonds and hydrophobic interactions. A lower binding value (kcal/mol) showed a lower energy of desolvation, indicating that the ligand–protein complex was more stable [48,49]. Based on the E-value against various protein targets, the order of WMT affinity was found to be caspase-3 > TRPV1 > p38 > P2Y > TRPM8 > JNK. Thus, we speculate that WMT is flexibly complexed with protein targets by establishing H-bonds and hydrophobic interactions and efficiently docked inside the active site of the target proteins.

## 4. Materials and Methods

### 4.1. Experimental Procedure

#### 4.1.1. Reagent and Chemicals

Withametelin was obtained from Asst. Prof. Ihsan Ul Haq (Quaid-i-Azam University, Islamabad). Vincristine was from Pharmedic Laboratories (Pvt) Ltd., Lahore, Pakistan; gabapentin from Lowitt Pharmaceutical (Pvt) Ltd., Peshawar, Pakistan; and 1-chloro-2,4-dinitrobenzene (CDNB), trichloroacetic acid (TCA), and 5-5’dithio-bis-2-nitro benzoic acid (DTNB) from Sigma-Aldrich (St. Louis, MO, USA). The following were purchased from suppliers: antibodies such as an anti-ERK, anti-JNK, anti-p38 MAPK, anti-TRPV1, anti-TRPM8, anti-P2Y, anti-caspase-3, anti-Bax, and ant-Bcl2 antibodies (Santa Cruz Biotechnology, Santa Cruz, CA, USA). Roswell Park Memorial Institute (RPMI) 1640 Medium was from Gibco, fetal bovine serum (FBS) from Invitrogen, San Diego, CA, USA, and penicillin/streptomycin from Gibco. Diaminobenzidine substrate (DAB) and MTT (3-(4,5-dimethylthiazol-2-yl)-2,5-diphenyl tetrazolium bromide) were from Sigma-Aldrich (St. Louis, MO, USA). All other chemicals used were of analytical grade.

#### 4.1.2. Experimental Animals

Male (Balb/c) mice weighing (25–30 g) were purchased from the National Institutes of Health (NIH) (Islamabad, Pakistan). All animal procedures were performed according to the protocols of the NIH Guidelines for the Care and Use of Laboratory Animals. The experimental protocols were reviewed and approved by Quaid-i-Azam University’s bioethical committee of research and animal use (Approval No: BEC-FBS-QAU2020-236, approved on 21 September 2020). The utmost care was taken to ensure that the animals were not harmed. All animal procedures were executed in the microbial-free zone laboratory of pharmacology. Mice were kept in controlled environmental conditions (21–25 °C, 55% ± 15% relative humidity) under a twelve-hour regular light/dark cycle and were given access to food and water ad libitum. An adequate quantity of wood shavings that covered the whole floor was provided as bedding.

#### 4.1.3. Induction of VCR-Induced Peripheral Neuropathy

Peripheral painful neuropathy was induced in mice by the administration of vincristine sulfate, according to the previously reported method [2,28]. Briefly, vincristine sulfate (75 μg kg, i.p.) was administered intraperitoneally once-daily for 10 consecutive days (from day 1–10). A treatment schedule is depicted in Figure 16.

### 4.2. Experimental Protocol

After 1 week of acclimatization, the mice were randomly divided into different groups (*n* = 10 mice/group). All efforts were made to minimize mouse suffering and the number of mice used.

Group I: Normal control

Mice were not subjected to vincristine administration and were kept for 21 days.

Group II: VCR

Vincristine (75 μg/kg, i.p.) was administered to mice for 10 days, and they were kept for 21 days.

Group III: GBP (60 mg/kg)

The mice were injected with VCR (75 μg/kg, i.p.) for 10 days. Then, gabapentin (60 mg/kg, i.p.) was injected for 11 consecutive days (from day 11–21) after the induction of neuropathy.

Group IV: WMT (0.1 mg/kg)

The mice received VCR in a dose of 75 μg/kg, i.p. for 10 days. Then, on the 11th day, withametelin (0.1 mg/kg, i.p.) was injected for 11 consecutive days (from day 11–21).

Group V: WMT (1 mg/kg)

The mice were injected with VCR (75 μg/kg) i.p. for 10 days. Then, withametelin (1 mg/kg) was administered i.p. for 11 consecutive days (from day 11–21) after the induction of neuropathy.

### 4.3. Pain Behavioral Tests

All behavioral testing was carried out by an experimenter blinded to the specific treatment groups. The pain behavioral tests used in the current study included mechanical allodynia, thermal hyperalgesia, and cold allodynia.

#### 4.3.1. Mechanical Allodynia

A set of calibrated Von Frey monofilaments were used to evaluate the mechanical allodynia. Mice were placed on an elevated mesh screen. Before the behavioral test, the mice were given a 20–30 min of acclimatization time. In order of increasing bending force, calibrated von Frey filaments were applied to the plantar surface of each hind paw. Each filament was applied 5 times to the plantar surface of each paw. The nocifensive responses such as paw withdrawal reflex in at least 3 of the 5 applications were defined as a positive response. The force at which the nocifensive response (paw withdrawal) occurred was recorded [50,51,52].

#### 4.3.2. Thermal Hyperalgesia

The threshold of withdrawal responses to noxious heat stimuli was measured as thermal hyperalgesia using a hot plate. Briefly, mice were individually put on a hot plate with the temperature set to 55 °C for this procedure. A 30-s cutoff time was used to prevent any paw or body damage. In the hot plate, the paw latency reaction (paw licking, flinching, or jumping) time was measured [53,54].

#### 4.3.3. Cold Allodynia

Cold hypersensitivity was assessed by measuring the responses to an acetone droplet on the hind paw. In the cold allodynia test, the mice were individually put on a wire mesh and were allowed to acclimatize. Then, a single drop (50 μL) of acetone was sprayed to the mid-plantar area of each hind paw in sequence without touching the skin. The paw withdrawal response (licking, withdrawal, and repeated and prolonged flinching of the paw) was recorded. Each paw was assessed once on the test day, with at least 5 min between tests. The two hind paws’ average withdrawal latency was then measured [37,50].

### 4.4. Cell Culture and Treatment

The pheochromocytoma (PC12) cell line was provided by Prof. Yeong Shik Kim (Emeritus Professor, SNU, Seoul, Korea). The neuroprotective activity of WMT was assessed in PC12 cells against H_2_O_2_-induced cytotoxicity, as previously described, with slight modifications. PC12 cells were maintained in RPMI 1640 supplemented with 10% FBS (*v*/*v*) and antibiotics (100 U/mL penicillin and 100 mg/mL streptomycin) at 37 °C with 5% CO_2_ in a humidified incubator. Briefly, cells were seeded into a 96-well culture plate at a density of 1 × 10^5^ cells/well and incubated for 24 h at 37 °C. All experiments were carried out 24 h after the cells were seeded. PC12 cells were pretreated with various doses of WMT (1, 10, 50, and 100 µM) for 2 h and then treated with 200-µM H_2_O_2_ for 24 h. Cells incubated only with H_2_O_2_ alone served as the H_2_O_2_ group, while untreated cells served as the control. WMT was dissolved in DMSO. The final concentration of DMSO was less than 0.1% (*v*/*v*).

### 4.5. MTT Assay

Cell viability was determined using an MTT assay, as described previously [55,56]. Briefly, PC12 cells were cultured in a 96-well plate (1 × 10^5^ cells/well) and incubated for 24 h at 37 °C. PC12 cells were pretreated with different concentrations of WMT before treatment with 200-µM H_2_O_2_. Then, 20-µL MTT solution was added (final concentration 5 mg/mL) and incubated at 37 °C for 4 h to produce formazan crystals. Subsequently, the supernatant was removed, and the formazan crystals were dissolved in dimethyl sulfoxide. The absorbance was measured using a spectrophotometric microtiter plate reader at 570 nm. The viability of the treated PC12 cells compared to the control cells was presented as a percentage (100%).

### 4.6. Tissue Preparation

For ex vivo assays, the animals were deeply anesthetized with ether, and blood samples were obtained from the heart of mice on the 21st day after the completion of the behavioral test. Serum was collected by centrifugation and preserved at −80 °C for biochemical analysis. After serum collection, anesthetized mice were killed through cervical dislocation, followed by the immediate isolation of the whole brain, spinal cord (L4–L6), and sciatic nerve. Until the assay, the tissue samples were kept at −80 °C. For the histopathological examinations, the samples of the brain and spinal cord were stored in the fixative brain solution (10% formalin). Correspondingly, the samples of the sciatic nerve were carefully dissected from the proximal aspect of the thigh to the knee joint proximal to its point of division into the common peroneal, tibial, and sural nerves. After dissecting the sciatic nerve, the middle part of the nerve was harvested and then stored in the fixative solution (10% formalin). Following fixation, the tissues were dehydrated using a graded ethanol series (from 70% to 100%). After dehydration, the tissues were cleaned with xylene. After clearing, the tissue sections were infiltrated with paraffin wax. After infiltration, the tissues were embedded into a paraffin wax block to enable sectioning on a microtome. The temperature of the embedded paraffin was kept 2–4 °C above the melting point of the wax. The paraffin-embedded tissues were sectioned into 5-µm thickness using a microtome.

### 4.7. Biochemical Parameters

After completion of the behavioral analysis, on day 21, the animals were deeply anesthetized, and the cervical dislocation procedure was used to sacrifice all of the groups of animals, followed by the immediate isolation of the brain and spinal cord. Similarly, the sciatic nerve was immediately obtained from the proximal aspect of the thigh to the knee joint proximal to its point of division into the common peroneal, tibial, and sural nerves. The brain was carefully dissected on an ice-chilled glass plate into the anterior cingulate cortex (ACC), insular cortex (IC), periaqueductal gray (PAG), hippocampus (HC), and amygdala. In the same way, the segment of the spinal cord (L4–L6) and sciatic nerve were also dissected on an ice-chilled glass plate. Parts of the freshly collected brain, spinal cord, and sciatic nerve were weighed, homogenized in prechilled phosphate-buffered saline (PBS; pH 7.5), and centrifuged at 1500× *g* for 10 min at 4 °C. The supernatant was used for the biochemical analysis.

#### 4.7.1. Reduced Glutathione

The reduced glutathione (GSH) levels were measured according to the previously described method [57,58]. The principle of this assay was based on the reaction of DTNB with the sulfhydryl group of GSH that resulted in the development of a yellow color. The reaction mixture contained 50 µL of supernatant, 0.05-M phosphate buffer (pH = 8), and the DTNB reagent. The spectrophotometric measurement of GSH’s conversion of DTNB to TNB was an indicator of the GSH content in the sample. The absorbance was measured with a spectrophotometer at a wavelength of 412 nm, and the results were expressed in percentages.

#### 4.7.2. Glutathione S-transferase

The Glutathione-S-transferase (GST) level was measured according to previously reported protocols [37]. The method used was based on the GST-catalyzed reaction between GSH and the GST substrate, such as CDNB. The glutathione-S-transferase activity mixture consisted of a phosphate buffer (0.1 mol, pH 6.5), reduced glutathione (1 mmol), CDNB (1 mmol), and 10–50-µL tissue homogenate. The changes in the absorbance were recorded at 340 nm by a spectrophotometer. The results were expressed in percentages.

#### 4.7.3. Catalase Activity

The catalase activity was determined according to the protocol reported previously [57]. The principle of this assay was based on the decomposition of H_2_O_2_ by catalase. The catalase activity mixture consisted of H_2_O_2_ (10 mmol/L) in a buffer containing 10–50 µL of tissue homogenate. A spectrophotometer was used to record changes in the absorbance at 240 nm. The results were shown in percentages.

#### 4.7.4. Superoxide Dismutase Assay

The level of superoxide dismutase SOD was measured according to the previously established method [59]. The reaction mixture consisted of pyrogallol (24 mM) and Tris-EDTA (50 mM), pH 8.5. An absorbance shift was recorded at 420 nm by a spectrophotometer.

#### 4.7.5. Estimation of Lipid Peroxidation

The lipid peroxidation assay was performed by measuring the MDA level according to the previously described protocol [60]. The quantitative measurement of malondialdehyde (MDA) in the tissue homogenate was an index of the lipid peroxidation. The reaction mixture included a phosphate buffer (0.1 mol; pH 7.4), tissue homogenate sample, ascorbic acid (100 mmol), and ferric chloride (100 mmol). For 1 h, the reaction mixture was incubated in a shaking water bath at 37 °C. The addition of 10% trichloroacetic acid stopped the reaction. Following the addition of 0.67% thiobarbituric acid, all tubes were put in a boiling water bath for 20 min before being transferred to a crushed ice bath and centrifuged for 10 min at 2500× *g*. The absorbance was recorded by a UV spectrophotometer at 532 nm, and the results were expressed in percentages.

#### 4.7.6. Myeloperoxidase Activity

The myeloperoxidase (MPO) activity was related to the neutrophilic infiltration into the site of inflammatory insult. The MPO activity was measured in the brain, spinal cord, and sciatic nerve by the previously established protocol [60,61]. The change in absorbance was recorded by a spectrophotometer at 470 nm. One unit of MPO activity was defined as the quantity catalyzing the decomposition of 1 µmol of hydrogen peroxide to water per min at 37 °C. The results of the MPO activity were expressed in percentages.

#### 4.7.7. Nitric Oxide

The production of nitric oxide (NO) was measured using the Griess reagent according to a previously reported method. Briefly, 50 µL of tissue homogenate and 50 µL of saline were carefully mixed with an equal amount of Griess reagent, then incubated at room temperature for 30 min. The absorbance was recorded at 560 nm using a microplate reader. A standard curve method developed by NaNO_2_ was used to calculate the NO. The results of the NO were expressed in percentages [53,62,63].

#### 4.7.8. Estimation of TNF-α and IL-1β

The brain (ACC, IC, PAG, HC, and amygdala); lumbar spinal cord (L4–L6); and sciatic nerve homogenates were centrifuged for 10 min at 10,000× *g* at 4 °C. The supernatants were immediately used for the measurement of the inflammatory cytokine (TNF-α and IL-1β) levels in triplicate using a mouse ELISA kit (Invitrogen, San Diego, CA, USA). The procedure was followed according to the manufacturer’s instructions [64].

### 4.8. Histopathological Analysis

#### 4.8.1. Hematoxylin and Eosin (H&E) Staining of the Brain, Spinal Cord, and Sciatic Nerve

The H and E staining of the vital organs, including the brain, spinal cord, and sciatic nerve, were performed to investigate the VCR-induced histopathological changes according to previously established protocols [65,66]. The brain, spinal cord, and sciatic nerve samples, intended for histopathological examination, were carefully removed and fixed immediately in 10% formalin. Following fixation, the wet fixed tissues were dehydrated using a graded ethanol series (from 70% to 100%). After dehydration, the tissues were cleaned with a clearing agent (xylene). After clearing, the tissue sections were infiltrated with paraffin wax. After infiltration, the tissues were embedded into a paraffin wax block to enable sectioning on a microtome. The temperature of the embedding paraffin was kept 2–4 °C above the melting point of the wax. The paraffin-embedded tissues were sectioned into 5-µm thickness using a microtome. Following sectioning, the tissue samples were placed on slides. Next, the tissue slides were deparaffinized using absolute xylene (100%), then rehydrated with absolute ethanol, gradient ethanolic concentrations (70–95%), and, finally, with distilled water. After that, the slides were washed in PBS and incubated with hematoxylin for 10 min. After that, the slides were placed in a glass jar under running tap water for 5 min. The slides were then examined under a microscope for nuclear staining, and if the staining was not clear, the hematoxylin timing was increased. The slides were then treated with 1% HCl and 1% ammonia water for a short interval and immediately rinsed with water again. These were then immersed in an eosin solution for 5–10 min, followed by rinsing with water and, finally, air-dried. The slides were then dehydrated using graded ethanol (70%, 95%, and 100%); fixed in xylene; and cover-slipped. The slides were analyzed under a microscope (Olympus, Corporate Parkway, Center Valley, PA, USA), and photos were taken at a magnification of ×4 and ×10 under a microscope.

#### 4.8.2. Masson’s Trichrome Staining

Briefly, tissue samples (sciatic nerves) were obtained and then fixed in 10% formalin at room temperature for 4 days and embedded in paraffin. Sections 5-µm thick were sliced from paraffin blocks using a microtome. Sections were stained with Masson’s Trichrome using the Masson Trichrome Stain kit-methyl/aniline blue per the manufacturer’s instructions before the morphological/morphometric examination. Morphometric analysis of the sciatic nerve sections, the myelin sheath thickness, the numbers of myelinated nerve fibers, and nerve fiber diameter was measured in five different areas from each section. All slides were photographed using a microscope (Olympus, Corporate Parkway, Center Valley, PA, USA) and photos were taken at a magnification (×10) [67].

### 4.9. Differential Scanning Calorimetry

DSC is a high-tech thermoanalytical tool that is used for the diagnosis of diseases and disease mechanisms at the molecular level. DSC measurements were taken using a Nano-DSC according to the manufacturer’s instructions as described previously [39]. Briefly, the device was preheated and balanced. Heating and cooling scans were performed at a scan rate of 1 K min^−1^ in the 20–110 °C range. Typically, for each sample, 2–4 heating-cooling cycles were performed. The thermal profile of native samples was seen in the first heating scans, while the second, third, and fourth heating scans showed practically identical profiles typical of denatured samples.

### 4.10. Fourier-Transform Infrared Spectroscopy

FTIR is a sensitive and highly reproducible physicochemical analytical technique that identifies structural moieties of biomolecules based on their IR absorption. Briefly, the sciatic nerve samples were lyophilized and used in an FTIR spectrometer. The samples were subjected to the range of 450 to 4000 cm^−1^ The FTIR spectrometer IR tracer (Shimadzu, Kyoto, Japan) was used to obtain the infrared spectra of sciatic nerve samples [37].

### 4.11. qRT-PCR Analysis of the TRPV1/TRPM8 and JNK

The quantitative RT-PCR was used to determine the effect of WMT on the mRNA expression levels of transient receptor potential (TRP) channels (TRPV1/TRPM8), purinergic receptor (P2Y), and JNK proteins in the spinal cord of VCR-treated mice. Trizol reagent was used to extract the mRNA from the lumbar dorsal spinal cord (L4–L6) tissue according to the manufacturer’s instructions. The mRNA expression levels of these proteins were measured using qRT-PCR as reported previously [37,61].

### 4.12. Single-Cell Gel Electrophoresis

Single-cell gel electrophoresis (comet assay) was carried out to measured DNA damage (genotoxicity) in the sciatic nerve as described previously [48,68,69,70]. Methanol-dipped slides were burnt over the flame to remove machine oil and dust. Then, the sterile slides (3/4 portion) were dipped in a normal melting point agarose solution (1% NMPA) and kept at a temperature of 25 °C. A small portion of the sciatic nerve was homogenized in the cold lysing solution (1 mL), then tissue homogenate and low melting point agarose (LMPA) was layered on NMPA coated slides, covered with a coverslip, and kept on ice for 10–12 min. The coverslips were removed after 10–12 min and treated again with the LMPA. After the third time coating with LMPA, the slides were dipped in lysing solution for 10 min followed by incubation for 2 h in the freezer. After electrophoresis, for the visualization of the DNA damage, the slides were stained with 1% ethidium bromide and examined under a fluorescent microscope. The software (CASP version 1.2.3.b, Krzysztof Ko´nca, CaspLab.com, accessed on 1 May 2021) was used to quantify the degree of DNA damage. The tail length and % DNA in the tail was used to measure the amount of DNA damage [49].

### 4.13. Immunohistochemical Analysis

Immunohistochemistry (IHC) was performed to investigate the effect of WMT on TRP channels (TRPV1/TRPM8), purinergic receptor (P2Y), MAPK (ERK/JNK/p38-MAPK), and apoptotic (Bax/Bcl-2/Casp-3) protein expression in the paraffin-embedded sections of the mouse spinal cord using the avidin–biotin–peroxidase complex (ABC) method, as described previously. Antibodies used in the IHC include primary antibodies (anti-ERK, anti-JNK, anti-p38 MAPK, anti-TRPV1, anti-TRPM8, anti-P2Y, anti-caspase-3, anti-Bax, and ant-Bcl2 antibodies) and biotinylated secondary antibody (Santa Cruz Biotechnology, Santa Cruz, CA, USA). Briefly, spinal cord transverse sections were incubated with different antibodies mentioned above and the reagents required for the ABC method were added. Immunoreactions were visualized using diaminobenzidine (DAB) for the detection of the antigen-antibody complex. The slides were analyzed under a microscope (Olympus, Center Valley, PA, USA) and immunohistochemical photos were taken. The immunoreactivity of the target proteins was quantified using ImageJ software 1.48 version (NIH, Rockville, MD, USA) (Java 1.8.9_66), according to the previously reported protocols [48,71].

### 4.14. Molecular Docking

The computational analysis was performed using AutoDock vina to assess the interaction of WMT and protein targets. The WMT was docked against the TRPV1 (PDB: 3J5R), TRPM8 (PDB ID: 6O77), P2Y (PDB ID: 4NTJ), JNK (PDB ID: 1UKI), p38 (PDB ID: 5OMH), and caspase-3 (PDB ID: 2J32). The proteins were downloaded from the protein data bank (PDB) and saved as a PDB file. The proteins were prepared in a discovery studio and actives sites were determined. The ligand, i.e., withametelin, was downloaded from PubChem and saved as an SDF file. The SDF file was then converted to the PDB in the discovery studio. The binding energies (kcal/mol) and amino acid residue involved in the hydrophobic interactions were analyzed.

### 4.15. Statistical Analysis

GraphPad Prism 8.0.2 (GraphPad Software Inc., San Diego, CA, USA) was used for the statistical analysis of data. The ImageJ software 1.48 version (NIH, Rockville, MD, USA) was used to analyze the morphological data. The software (CASP version 1.2.3.b, Krzysztof Ko´nca, CaspLab.com, accessed on 1 May 2021) was used to quantify the degree of DNA damage. Data from behavior and biochemical analyses were expressed as mean ± standard deviations (SD). The Normality and equality of variance were confirmed using Shapiro-Wilk’s and Brown Forsythe’s tests, respectively. Data from behavioral and biochemical were analyzed using one-way analysis of variance (one-way ANOVA) followed by Bonferroni’s post hoc test. In all calculations, *p* < 0.05 was considered to be statistically significant.

## 5. Conclusions

In conclusion, the present study provided convincing evidence that the withametelin treatment significantly alleviated VCR-induced neuropathic pain based on the results obtained from the behavioral, biochemical, histopathological, and computational data. The molecular mechanisms of the current study were attributed to the suppression of the TRPV1/TRPM8/P2Y nociceptors and MAPK signaling in mice (Figure 17). Thus, based on the present findings, it was concluded that WMT is a potential novel antinociceptive candidate for neuropathic pain.

## Figures and Tables

**Figure 1 ijms-22-06084-f001:**
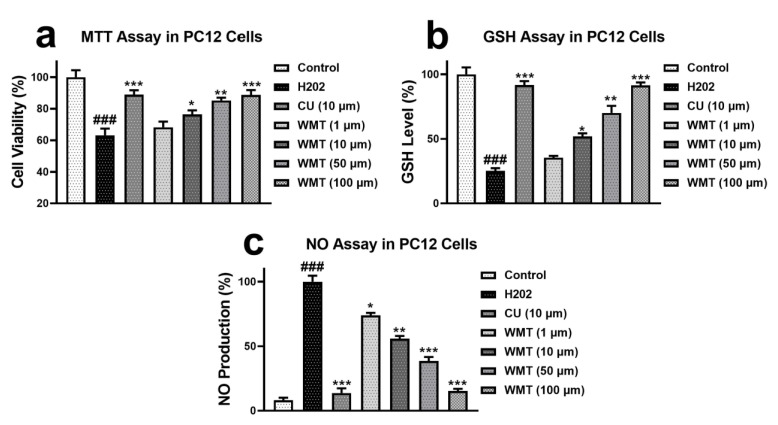
Effect of WMT (1, 10, 50, and 100 μM) on the cell viability (**a**), GSH level (**b**), and NO production (**c**) in H_2_O_2_-induced PC12 cells. Cell viability was determined by the MTT assay. Data were obtained from the three sets of independent experiments and were expressed as the mean ± S.D. *n* = 3 independent experiments; each independent experiment consisted of 5 replicates. (*) *p* < 0.05, (**) *p* < 0.01, and (***) *p <* 0.001 indicates significant differences from the H_2_O_2_-treated group. (###) indicates a significant difference from the control group.

**Figure 2 ijms-22-06084-f002:**
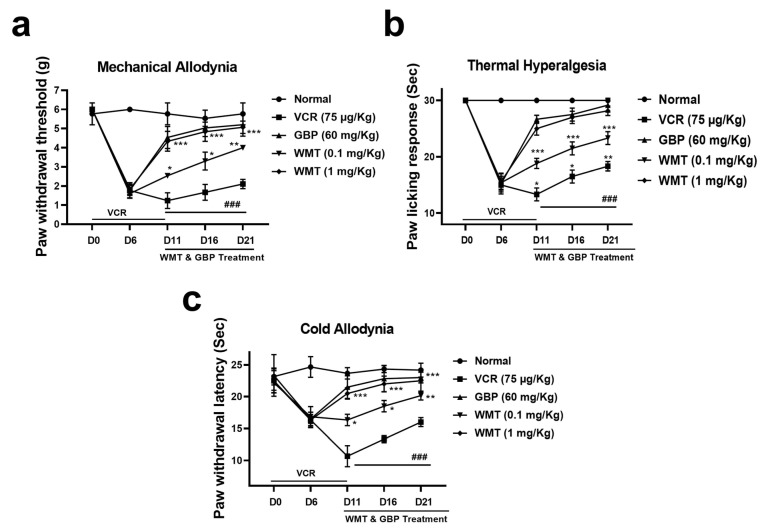
Effects of withametelin (0.1, 1 mg/kg) and GBP (60 mg/kg) on VCR-induced mechanical allodynia, (**a**) thermal hyperalgesia, (**b**) and cold allodynia (**c**). All results were expressed as the mean (*n* = 10) ± SD. **###**
*p* < 0.001 compared to the normal control group; * *p* < 0.05, ** *p* < 0.01, and *** *p* < 0.001 compared to the VCR group.

**Figure 3 ijms-22-06084-f003:**
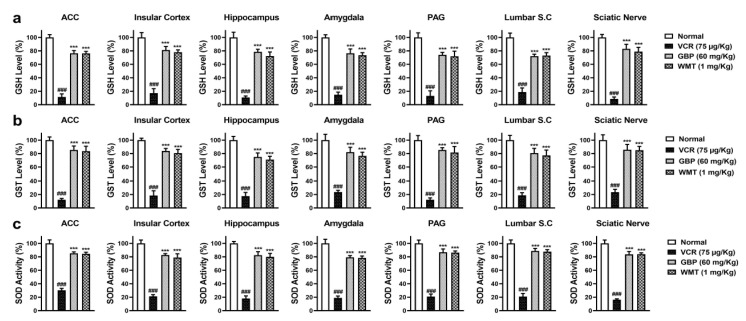
The effect of withametelin on the antioxidant levels in the different regions of the brain, spinal cord, and sciatic nerve of the VCR model. GSH (**a**), GST (**b**), and SOD levels (**c**). The results were expressed in percentages. All results were expressed as the mean (*n* = 5) ± SD. The experiment was performed in triplicate independently. ### *p* < 0.001 compared to the normal control group; *** *p* < 0.001 compared to the VCR group.

**Figure 4 ijms-22-06084-f004:**
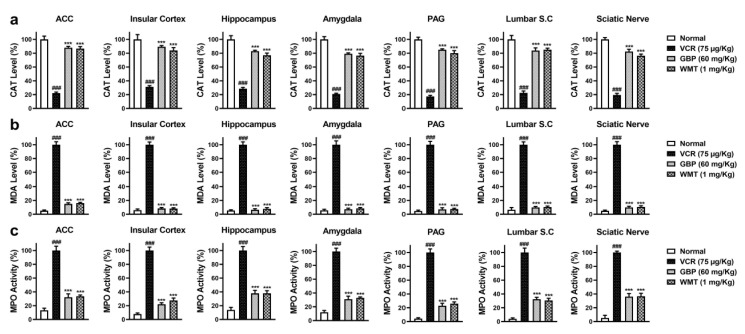
The effect of withametelin on the levels of the antioxidants and oxidative markers in the different regions of the brain, spinal cord, and sciatic nerve of the VCR model. Catalase (**a**), MDA (**b**), and MPO levels (**c**). The results were expressed in percentages. All results were expressed as the mean (*n* = 5) ± SD. The experiment was performed in triplicate independently. ### *p* < 0.001 compared to the normal control group; *** *p* < 0.001 compared to the VCR group.

**Figure 5 ijms-22-06084-f005:**
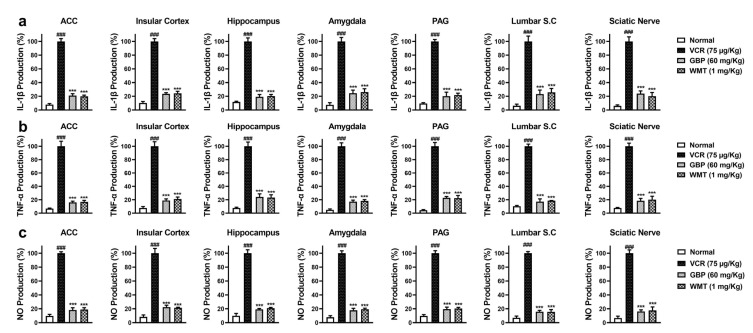
The effect of withametelin on the levels of the cytokines and NO production in the different regions of the brain, spinal cord, and sciatic nerve of the VCR model. IL-1β (**a**), TNF-α (**b**), and NO production (**c**). The results were expressed in percentages. All results were expressed as the mean (*n* = 5) ± SD. The experiment was performed in triplicate independently. ### *p* < 0.001 compared to the normal control group; *** *p* < 0.001 compared to the VCR group.

**Figure 6 ijms-22-06084-f006:**
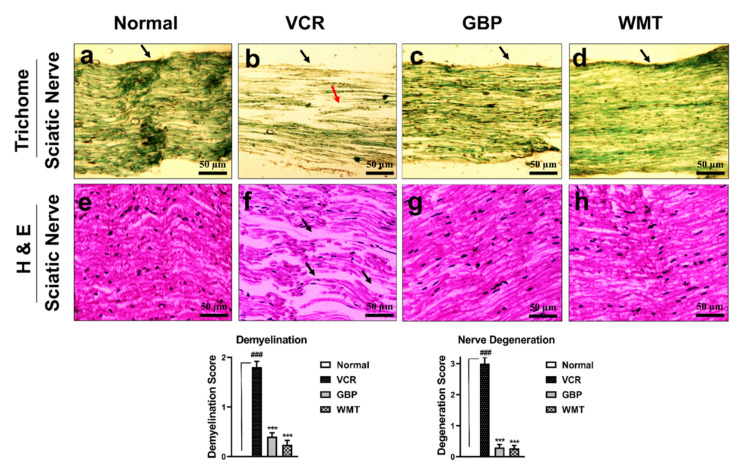
Effect of withametelin on the histopathological changes in the sciatic nerve (H&E and Trichome staining 10×). Representative photomicrographs of the sciatic nerve. Normal (**a**,**e**), VCR (**b**,**f**), GBP (**c**,**g**), and WMT (**d**,**h**). WMT and GBP inhibited histopathological alterations in the sciatic nerve. In (**a**), the black arrowhead shows a normal myelin sheath and normal fiber arrangement. In (**b**), the black arrowhead shows demyelination, and the red arrowhead shows fiber derangement. While (**c**,**d**) show recovery in the fiber derangement and demyelination. Similarly, in (**f**), the black arrowhead shows nerve derangement/degeneration and axonal swelling. While (**g**,**h**) show recovery in the nerve derangement/degeneration and axonal swelling. All results were expressed as the mean (*n* = 3) ± SD. ### *p* < 0.001 compared to the normal control group; *** *p* < 0.001 compared to the VCR group.

**Figure 7 ijms-22-06084-f007:**
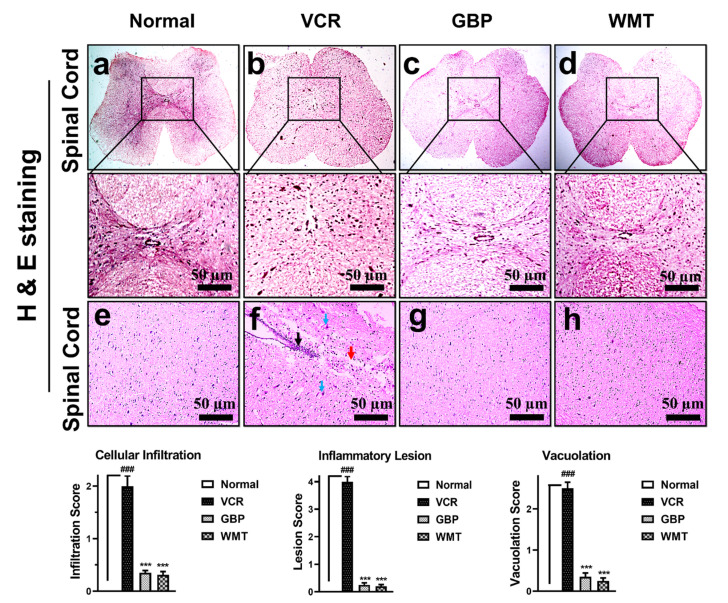
Effect of withametelin on the histopathological changes in the spinal cord. Representative photomicrographs of the transverse section of the spinal cord (H&E; 4× and 10×). Normal (**a**), VCR (**b**), GBP (**c**), and WMT (**d**). WMT and GBP suppressed the VCR-induced histopathological changes, including cellular infiltration/inflammatory lesion and vacuolation in the spinal cord. Representative photomicrographs of the longitudinal section spinal cord (H&E; 10×). Normal (**e**), VCR (**f**), GBP (**g**), and WMT (**h**). WMT and GBP ameliorated histopathological alterations in the spinal cord. The black arrowhead shows inflammatory lesions, the red arrowhead shows vacuolation, and the sky-blue arrowhead shows vacuolar changes in the neurons. All results were expressed as the mean (*n* = 3) ± SD. ### *p* < 0.001 compared to the normal control group; *** *p* < 0.001 compared to the VCR group.

**Figure 8 ijms-22-06084-f008:**
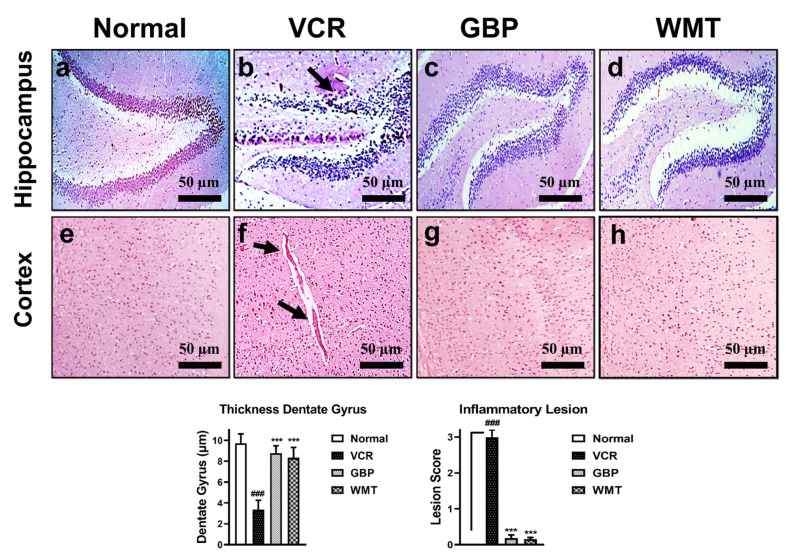
Effect of withametelin on the histopathological changes in the brain (hippocampus and cortex) (H&E; 10×). Representative photomicrographs of the dentate gyrus of the hippocampus. Normal (**a**), VCR (**b**), GBP (**c**), and WMT (**d**). WMT and GBP suppressed the reduction in the granular layer of the hippocampal dentate gyrus. The black arrowhead shows a reduction in the hippocampal dentate gyrus. Representative photomicrographs of the cortex. Normal (**e**), VCR (**f**), GBP (**g**), and WMT (**h**). The WMT treatment significantly reduced the neuropathology associated with VCR (inflammatory plaque formation in the cerebral cortex). The black arrowhead shows inflammatory plaque in the cortex. All results were expressed as the mean (*n* = 3) ± SD. ### *p* < 0.001 compared to the normal control group; *** *p* < 0.001 compared to the VCR group.

**Figure 9 ijms-22-06084-f009:**
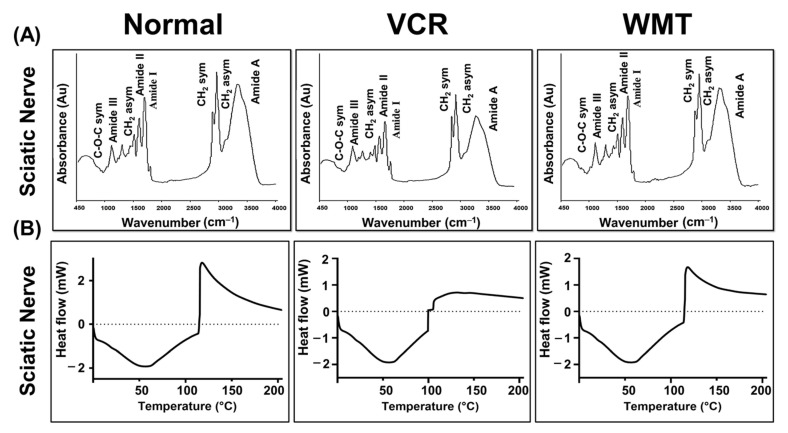
Representative FTIR spectra of the sciatic nerve of the normal, VCR, and withametelin groups (**A**). WMT significantly inhibited VCR-induced changes in the biochemical composition of the myelin sheath. Effect of the withametelin treatment on the protein structure (**B**). DSC thermogram confirmed that the withametelin treatment markedly suppressed any VCR-induced protein structure alterations.

**Figure 10 ijms-22-06084-f010:**
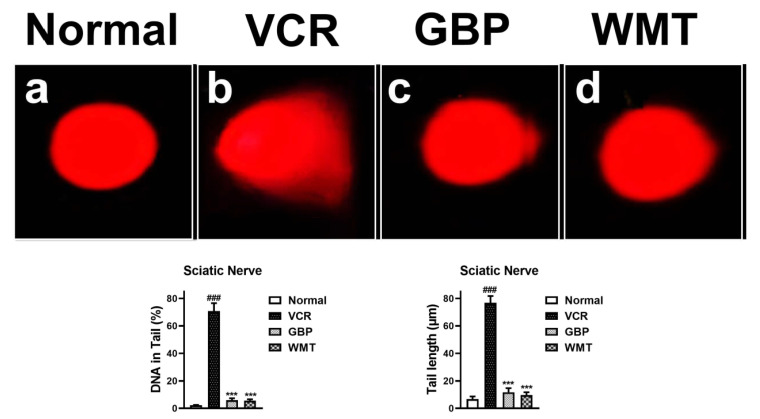
Effect of the WMT treatment against VCR-induced DNA damage of the sciatic nerve. Normal (**a**), VCR (**b**), GBP (**c**), and WMT (**d**). All results were expressed as the mean (*n* = 3) ± SD. ### *p* < 0.001 compared to the normal control group; *** *p* < 0.001 compared to the VCR group.

**Figure 11 ijms-22-06084-f011:**
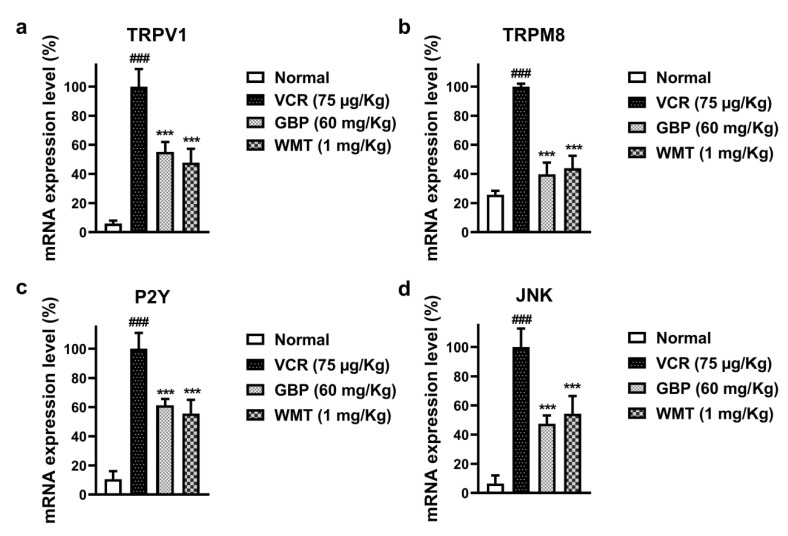
Effect of the WMT treatment on the TRPV1, TRPM8, P2Y, and JNK mRNA expression levels following VCR administration. TRPV1 (**a**), TRPM8 (**b**), P2Y (**c**), and JNK (**d**). The WMT treatment markedly reduced the mRNA expression level of the TRPV1, TRPM8, P2Y, and JNK proteins compared to the VCR group. All the results were expressed as the mean (*n* = 5) ± SD. The experiment was performed in triplicate independently. ### *p* < 0.001 compared to the normal control group; *** *p* < 0.001 compared to the VCR group.

**Figure 12 ijms-22-06084-f012:**
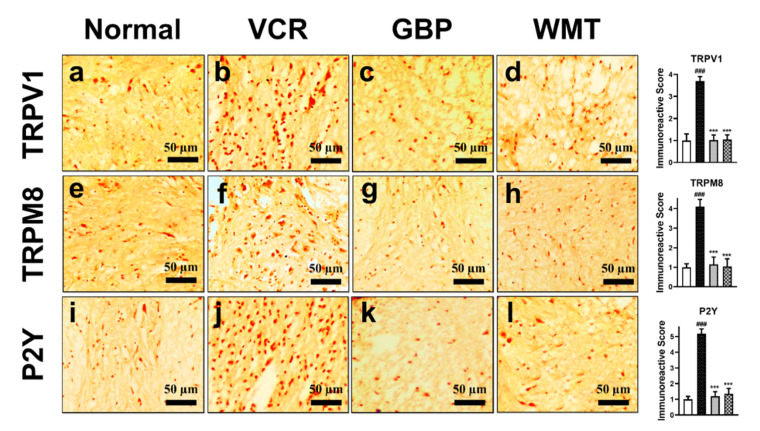
Effect of the withametelin treatment on the expression levels of the TRPV1/TRPM8/P2Y proteins in the mouse spinal cords. Magnification 10×. TRPV1 (**a**–**d**), TRPM8 (**e**–**h**), and P2Y (**i**–**l**). All the results were expressed as the mean (*n* = 3) ± SD. ### *p* < 0.001 vs. the control group. *** *p* < 0.001 vs. the VCR group.

**Figure 13 ijms-22-06084-f013:**
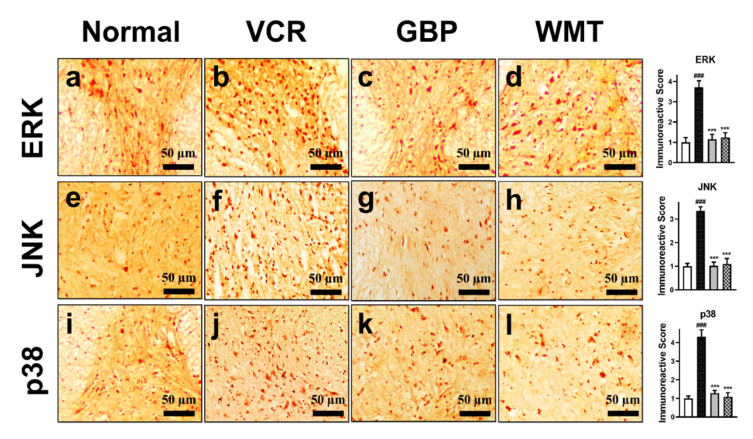
Effect of the withametelin treatment on the expression levels of the ERK/JNK/p38 proteins in the mouse spinal cords. Magnification 10×. ERK (**a**–**d**), JNK (**e**–**h**), and p38 (**i**–**l**). All the results were expressed as the mean (*n* = 3) ± SD. ### *p* < 0.001 vs. the control group. *** *p* < 0.001 vs. the VCR group.

**Figure 14 ijms-22-06084-f014:**
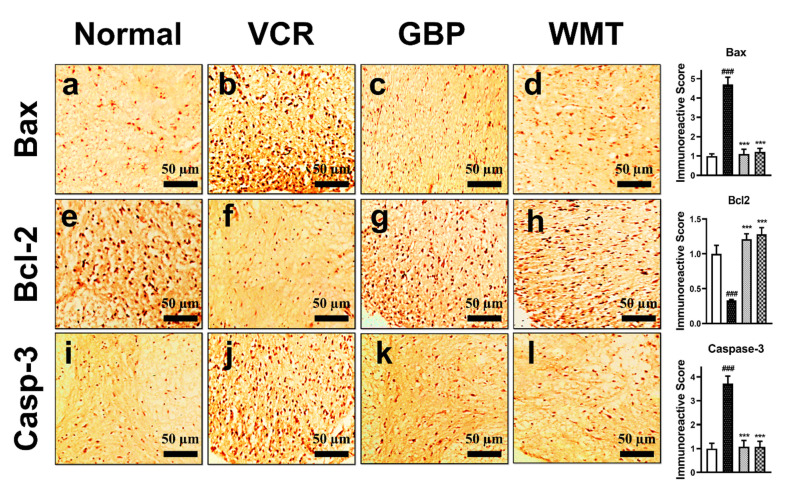
Effect of the withametelin treatment on the expression levels of Bax, Bcl-2, and caspase-3 proteins in the mouse spinal cords. Magnification 10×. Bax (**a**–**d**), Bcl-2 (**e**–**h**), and caspase-3 (**i**–**l**). All the results were expressed as the mean (*n* = 3) ± SD. ### *p* < 0.001 vs. the control group. *** *p* < 0.001 vs. the VCR group.

**Figure 15 ijms-22-06084-f015:**
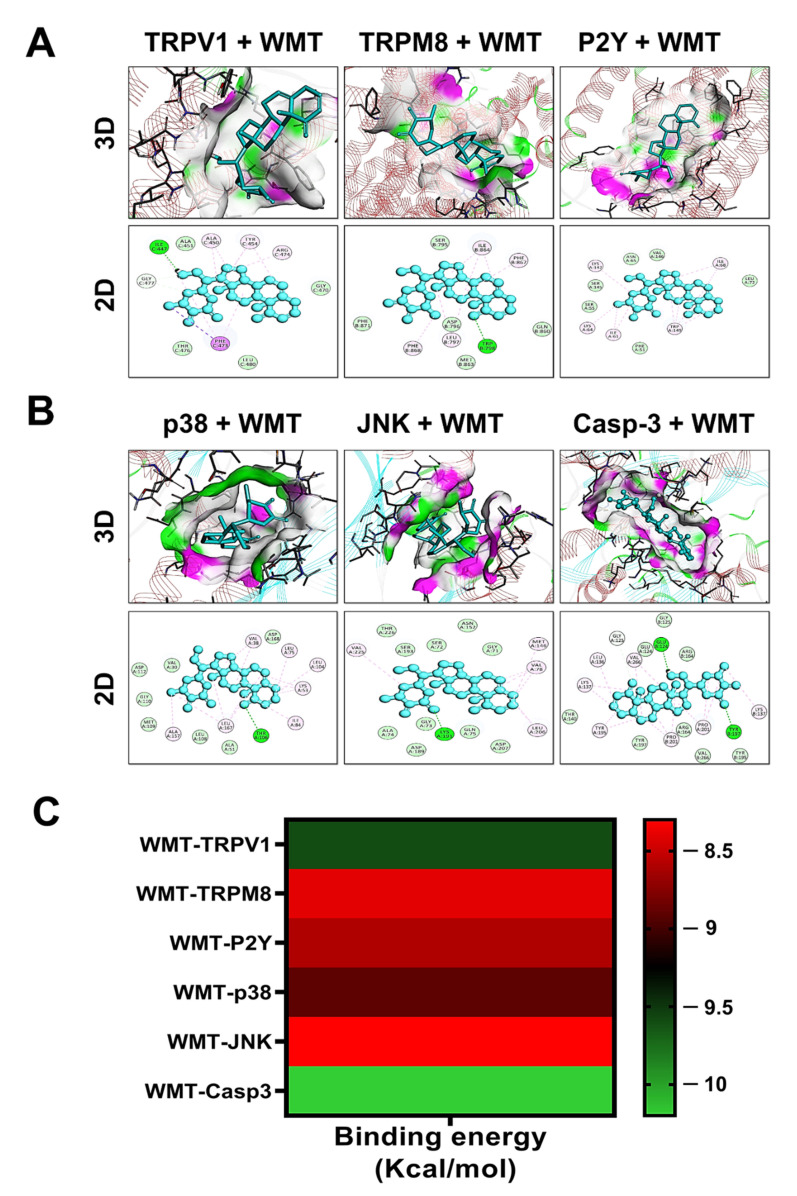
Docking analysis of withametelin with TRPV1, TRPM8, P2Y (**A**), p38, JNK, and caspase-3 (**B**). The two-dimensional (2D) and three dimensional (3D) structural interactions between the ligand and protein macromolecules are shown. Withametelin docked well in the active site of the protein targets. The heat map showed the binding energy of the protein targets (**C**). Withametelin exhibited a higher binding affinity with the protein targets and had multi-target binding efficiency.

**Figure 16 ijms-22-06084-f016:**
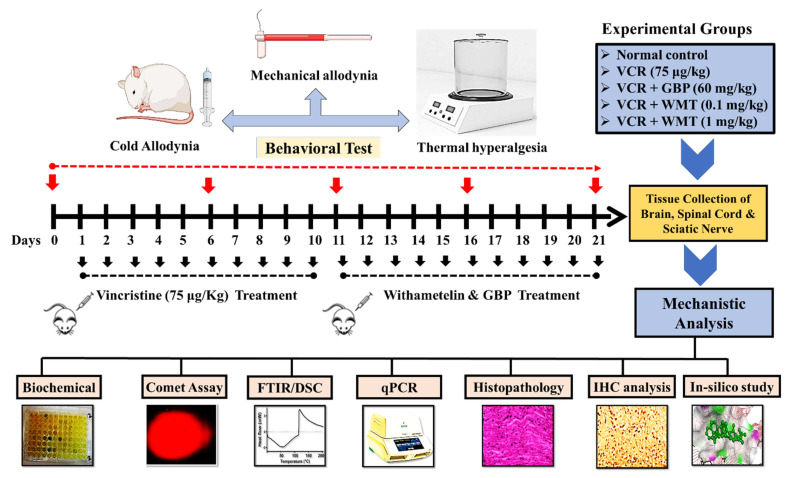
A schematic representation of the in vivo study design.

**Figure 17 ijms-22-06084-f017:**
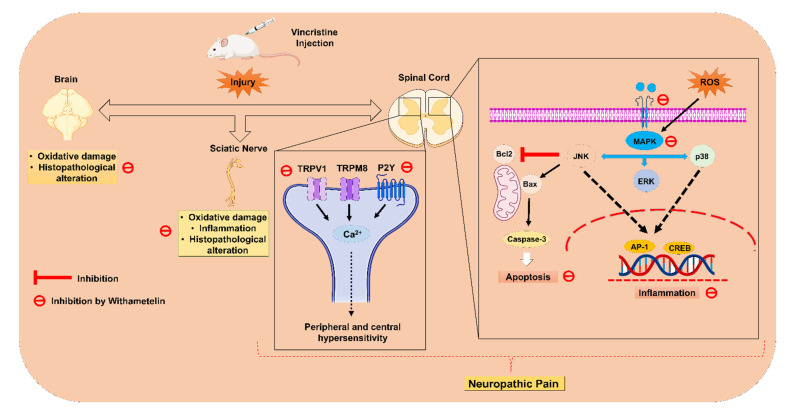
The proposed molecular mechanism of withametelin in the vincristine model of neuropathic pain. Created with BioRender.com, accessed on 1 May 2021.

**Table 1 ijms-22-06084-t001:** Spectral assignment of the band frequencies with a spectral range of 450–4000 cm^−1^.

Wavenumber (cm^−1^)	Spectral Assignment
3285	N-H str of amide A: Protein
2926	CH_2_ asym str: Lipids (fatty acid)
2850	CH_2_ sym str: Membrane lipids (membrane fatty acid)
1650	C=O sym str band of α-helical structure (amide I): Protein
1545	N-H bend, vib; C-N str of the amino acid (amide II): Protein
1460	CH_2_ sciss, vib; CH_2_ asym bend and vib of phospholipid: Membrane lipids
1300	C-N str, N-H bending; C=O str; O=C-N bend and vib (amide III): Protein
1062	Ester C-O-C sym str (phospholipids); ribose C-O str (Nucleic acids)

str, stretching; asym, asymmetric; sym, symmetric; bend, bending; vib, vibration; sciss, scissoring.

**Table 2 ijms-22-06084-t002:** The sequences of the forward and reverse primers.

Genes	Forward Primers	Reverse Primers
TRPV1	AAGGCTCTATGATCGCAGGA	CAGATTGAGCATGGCTTTGA
TRPM8	ACATACCAAGGAGTTTCCAACAG	GCTGGGTCAGCAGTTCGTAG
P2Y	CGTGCTGGTGTGGCTCATT	GGACCCCGGTACCTGAGTAGA
JNK	AGCCTTGTCCTTCGTGTC	AAAGTGGTCAACAGAGCC
β-actin	CATCACCATCGGAATGAG	CACGGTGTTGGCATACAGG

**Table 3 ijms-22-06084-t003:** The docking analysis of withametelin with TRPV1, TRPM8, P2Y, p38, JNK, and caspase-3.

WMT–Protein Interaction	Binding Energy (Kcal/mol)	Hydrogen Bond	Hydrophobic Interactions
WMT–TRPV1	−9.6	ILE C:447	ALA C:451, TYR C 454, ARG C:474, GLY C:477
WMT–TRPM8	−8.4	TRP B: 78	ILE B:864, PHE B:867, ASP B: 796, PHE B:868
WMT–P2Y	−8.6	----------	ILE A:68, LEU A:72, TRP A: 149, LYS A: 142
WMT–p38	−8.9	THR A:106	VAL A:38, LEU A:75, LYS A:53, ALA A:157
WMT–JNK	−8.3	LYS A:191	VAL A:78, LEU A: 206, VAL A:225, MET A:146
WMT–Casp-3	−10.2	GLU B:124 TYR B:197	VAL A:266, LYS A:137, TYR A:195, PRO B: 201

## Data Availability

The data presented in this study are available from the corresponding author upon reasonable request.

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
