# Peer review of "Suppression of TRPV1/TRPM8/P2Y Nociceptors by Withametelin via Downregulating MAPK Signaling in Mouse Model of Vincristine-Induced Neuropathic Pain"

_ijms, 2021, doi:10.3390/ijms22116084_

Round 1
Reviewer 1 Report
The paper entitled Suppression of TRPV1/TRPM8/P2Y nociceptors by 2 withametelin via downregulating MAPK signaling in mouse model of vincristine-induced neuropathic pain presented by Khan et al. Vincristine is widely used drug in the treatment of various types of cancer such as lymphomas, leukemia, breast cancer, and brain tumor. Authors are stressed the fact that therapeutic strategies used against VCR-induced neuropathic pain are limited to anti-convulsant, opioids, and tricyclic anti-depressants. WMT against VCR-induced apoptosis. By the way , paper needs some improvement.
I recommend to apply qPCR and western blot to analyze JNK , TRPV1, TRPV8 content in tissues and WMT against VCR-induced apoptosis .
Also, TRPV interactions withcould be proved by ChIP.
Author Response
Response: The reviewer is highly acknowledged for the valuable feedback. The feedback was very insightful and valuable for improving the quality of our paper. The antinociceptive effect of withametelin on JNK and TRPV1/TRPM8/P2Y protein expression has been already confirmed using immunohistochemistry. To further explore more in-depth, the mRNA levels of JNK and TRPV1/TRPM8/P2Y were investigated in the spinal cord using qPCR as per the reviewer's recommendation. Our results indicated that the mRNA expression levels of TRPV1/TRPM8/P2Y and JNK were significantly increased in the spinal cord of the VCR group. However, WMT treatment strikingly reduced mRNA expression levels of TRPV1/TRPM8/P2Y and JNK in the spinal cord of VCR-treated mice as shown in figure 11 of the revised manuscript.
Reviewer 2 Report
In the manuscript, entitled "Suppression of TRPV1/TRPM8/P2Y nociceptors by withametelin via downregulating MAPK signaling in mouse model of vincristine-induced neuropathic pain" Authors present a well thought out, complex study showing the promising analgesic properties of withametelin.
Although the study is very interesting, I have several major concerns about the manuscript, especially the Methods section should be improved in my opinion.
My comments are as follows:
Major comments:
- My biggest concern would be the lack of negative control (vehicle treatment) groups from all the experiments. It seems tissues were collected 10 days after vincristine treatment, therefore based on the presented data we can not exclude the effects of spontaneous recovering. If a negative control was applied it should be included in the Figures and the protocol (Fig. 17 as well). Kindly clear this issue.
- On the legends of Fig 3 and 4, does n=5 equal biological replicates, or the number of samples? Please include the number of biological replicates in the description. This should apply to all measurements.
- The legends on Fig. 7 are incorrect or mixed (GBP/VCR), please fix the legends. Again, kindly describe the results in more detail and show the values as an additional panel in the figure. The same for section 2.7 and Fig. 8.
- In the Discussion, Authors state: “results of the current study demonstrated that WMT showed a high binding affinity with the mentioned protein targets”, whereas there are apparent differences in the binding affinities among the investigated proteins. This should be addressed in the Discussion.
- Methods:
- Fig. 15 does not show clearly the days of behavioral analyses.
- In section 4.7 Authors state: “Cell viability was determined using an MTT assay as described previously”, but there is no study cited. Nevertheless, the method should be briefly described. The same applies to section 4.9.1.
- In section 4.8. the thickness of the sections is not given. What were the tissues kept in before and after sectioning? Only formalin solution was used? What part/which branch of the sciatic nerve was sectioned? What were the parameters (temperature etc.) during sectioning? What instrument was used for the process?
- Include a detailed description of the staining process as well (solutions, dilutions, etc.).
- The preparation of tissues/samples is missing from all the subchapters of section 4.9.
- Add a brief description of the method in section 4.11. Same for 4.14.
- Were the in vitro measurements, especially the scoring, performed in a blinded fashion?
- The Institutional Review Board Statement is missing.
- In the 2.5 Chapter of the Results section, Authors state that “WMT markedly attenuated (P < 0.001) vincristine-induced histopathological changes in the sciatic nerve”. Please include the value itself that was shown to be different. Also, show that result in Fig. 6 as an additional panel.
Minor comments:
- Most of the abbreviations are not described before they are introduced (e.g. GBP, WTT, WMT, etc.).
- I would indicate on Figures 3 and 4 the fact that co-treatments were applied.
- Please include the scale bars in the histological pictures.
- Please indicate the sample numbers on each Figure.
- The PC12 cell line consists of rat adrenal gland tissue cells. Why did the Authors choose this particular cell line for the investigation of neuronal protective effects?
- The Discussion should summarize the findings of the study with the proposition of a possible mechanism of action, connecting the different results together.
- Authors could refer back to the given Figures in the Discussion.
In summary, the manuscript presents the interesting results of a complex study. However, there are some parts that should be improved in my opinion, and the Methods section is not appropriate in its present form.
Author Response
Reviewer 2
In the manuscript, entitled "Suppression of TRPV1/TRPM8/P2Y nociceptors by withametelin via downregulating MAPK signaling in mouse model of vincristine-induced neuropathic pain" Authors present a well thought out, complex study showing the promising analgesic properties of withametelin.
Although the study is very interesting, I have several major concerns about the manuscript, especially the Methods section should be improved in my opinion.
My comments are as follows:
Major comments:
- My biggest concern would be the lack of negative control (vehicle treatment) groups from all the experiments. It seems tissues were collected 10 days after vincristine treatment, therefore based on the presented data we can not exclude the effects of spontaneous recovering. If a negative control was applied it should be included in the Figures and the protocol (Fig. 17 as well). Kindly clear this issue.
Response: We are very grateful to the reviewers for the valuable feedback. We express regret for not describing the study design in the method section. Indeed this was 21 days study and all experimental groups were observed for nociceptive behavior until 21 days according to the previously reported method [1, 2]. This study includes five experimental groups such as normal control, negative (VCR), positive (gabapentin), withametelin (0.1), and withametelin (1). For induction of peripheral neuropathy, vincristine (75 μg/kg) was administered intraperitoneally (i.p.) for 10 consecutive days (day 1-10) to all experimental groups except the normal group which did not receive any treatment. All experimental groups were kept for 21 days. It is well recognized that 11 consecutive doses of vincristine produced painful neuropathy which persists for 21 days [1, 2]. After induction of neuropathy on day 11, the posttreatment has been started. Gabapentin (GBP) (60 mg/kg, i.p.) and withametelin (0.1 and 1 mg/kg i.p.) treatments were given after the completion of VCR injection on the 11th day up to 21 days. On day 21 tissues were collected from all experimental groups for biochemical and histopathological analysis.
- On the legends of Fig 3 and 4, does n=5 equal biological replicates, or the number of samples? Please include the number of biological replicates in the description. This should apply to all measurements.
Response: As per the reviewer's suggestion the number of samples and biological replicates has been added and highlighted in the figure caption of the revised manuscript.
- The legends on Fig. 7 are incorrect or mixed (GBP/VCR), please fix the legends. Again, kindly describe the results in more detail and show the values as an additional panel in the figure. The same for section 2.7 and Fig. 8.
Response: The correction has been made in Fig. 7. The results in Fig. 7, 8 and section 2.5, 2.6, 2.7 has been described in detail in the revised manuscript.
- In the Discussion, Authors state: “results of the current study demonstrated that WMT showed a high binding affinity with the mentioned protein targets”, whereas there are apparent differences in the binding affinities among the investigated proteins. This should be addressed in the Discussion.
Response: As per the reviewer's recommendation the results of the docking analysis have been described in the discussion section of the revised manuscript.
- Methods:
- 15 does not show clearly the days of behavioral analyses.
Response: Fig. 16 has been modified and days of behavioral analysis have been added.
- In section 4.7 Authors state: “Cell viability was determined using an MTT assay as described previously”, but there is no study cited. Nevertheless, the method should be briefly described. The same applies to section 4.9.1.
Response: The method for MTT assay has been described and cited in section 4.7 of the revised manuscript. In the same way, the method has been briefly described and cited in section 4.9.1 as per the reviewer's recommendation.
- In section 4.8. the thickness of the sections is not given. What were the tissues kept in before and after sectioning? Only formalin solution was used? What part/which branch of the sciatic nerve was sectioned? What were the parameters (temperature etc.) during sectioning? What instrument was used for the process?
Response: As per the reviewer’s kind suggestion detailed description of histopathological analysis including tissue processing and H & E staining has been added in the 4.8 and 4.10.1 sections of the revised manuscript.
- Include a detailed description of the staining process as well (solutions, dilutions, etc.).
Response: The detailed description of staining processes has been added in the 4.10.1 section of the revised manuscript.
The preparation of tissues/samples is missing from all the subchapters of section 4.9.
Response: The preparation of tissue samples has been described and highlighted in section 4.9 of the revised manuscript.
- Add a brief description of the method in section 4.11. Same for 4.14.
Response: A brief description of the method for sections 4.11 and 4.14 has been added in the revised manuscript.
- Were the in vitromeasurements, especially the scoring, performed in a blinded fashion?
Response: Yes, to avoid experimenter bias the in vitro measurements were performed and evaluated in a blinded fashion.
- The Institutional Review Board Statement is missing.
Response: The Institutional Review Board Statement has been provided in the revised manuscript.
- In the 2.5 Chapter of the Results section, Authors state that “WMT markedly attenuated (P < 0.001) vincristine-induced histopathological changes in the sciatic nerve”. Please include the value itself that was shown to be different. Also, show that result in Fig. 6 as an additional panel.
Response: The results in fig. 6 have been revised as per reviewer's suggestion.
Minor comments:
- Most of the abbreviations are not described before they are introduced (e.g. GBP, WTT, WMT, etc.).
Response: As per suggestion all abbreviations are described and highlighted in the revised manuscript.
- I would indicate on Figures 3 and 4 the fact that co-treatments were applied.
Response: In this study, we evaluate the effect of WMT after induction of neuropathic pain. It is also biologically and therapeutically relevant since patients are to be treated with WMT in the future after the onset of symptoms.
- Please include the scale bars in the histological pictures.
Response: The scale bar has been added in the histological pictures of the revised manuscript.
- Please indicate the sample numbers on each Figure.
Response: The sample number has been added to each figure of the revised manuscript.
- The PC12 cell line consists of rat adrenal gland tissue cells. Why did the Authors choose this particular cell line for the investigation of neuronal protective effects?
Response: Thank you for raising an important point. The PC12 cell line is one of the most used cell lines in neuroscience research, including studies on neurotoxicity, neuroprotection, neurosecretion, neuroinflammation, and synaptogenesis [3]. Although these cell lines originate from a pheochromocytoma of the rat adrenal medulla, they have been extensively characterized for neurosecretion and exhibit features of mature neurons [4, 5]. The popularity of PC12 cells is mainly due to their extreme versatility for pharmacological manipulation, ease of culture, and a large amount of background knowledge on their proliferation and differentiation [3].
- The Discussion should summarize the findings of the study with the proposition of a possible mechanism of action, connecting the different results together. Authors could refer back to the given Figures in the Discussion.
Response: As per the reviewer's suggestion the discussion has been revised thoroughly.
In summary, the manuscript presents the interesting results of a complex study. However, there are some parts that should be improved in my opinion, and the Methods section is not appropriate in its present form.
Response: We are very grateful to the reviewer for the very thoughtful critique of our manuscript and for the valuable feedback. The feedback was very insightful and valuable for improving the quality of our paper. The method section has been improved in the revised manuscript as per reviewer's suggestion.
- Gong, S.-S., et al., Neuroprotective effect of matrine in mouse model of vincristine-induced neuropathic pain. 2016. 41(11): p. 3147-3159.
- Vashistha, B., A. Sharma, and V.J.N.n. Jain, Ameliorative potential of ferulic acid in vincristine-induced painful neuropathy in rats: an evidence of behavioral and biochemical examination. 2017. 20(1): p. 60-70.
- Wiatrak, B., et al., PC12 cell line: cell types, coating of culture vessels, differentiation and other culture conditions. 2020. 9(4): p. 958.
- Wang, W.-L., et al., Current situation of PC12 cell use in neuronal injury study. 2015. 4(2): p. 61-66.
- Westerink, R. and A.G.J.A.P. Ewing, The PC12 cell as model for neurosecretion. 2008. 192(2): p. 273-285.
Round 2
Reviewer 1 Report
Everything is OK. Authors addressed all my questions.
Author Response
NA
Reviewer 2 Report
I appreciate the Authors for addressing my comments, I believe that the manuscript has been approved substantially. I have a few comments remaining.
- In some of the figure legends, what do Authors mean by 'individual groups of mice'? n= 3 represents 3 biological replicates or the number of samples measured?
- In chapter 2.4 how were the differences shown? Although Authors refer to some observations, no value is given supporting these observations, only representative images.
- The same is true for chapters 2.5, and 2.6. For example: "VCR significantly reduced the granular layer of the hippocampal dentate gyrus." How was the significant difference determined? It should be presented as well, similarly to Fig. 13 or 14.
Author Response
Point to point comments and response
Reviewer 2
I appreciate the Authors for addressing my comments, I believe that the manuscript has been approved substantially. I have a few comments remaining.
- In some of the figure legends, what do Authors mean by 'individual groups of mice'? n= 3 represents 3 biological replicates or the number of samples measured?
Response: We are very grateful to the reviewer for encouraging appraisal and providing valuable feedback. Individual group of mice here means the number of mice used for histopathological analysis per group, in fact, “n” indicates the number of samples measured. As per the reviewer's suggestion, the caption has been modified in the revised manuscript.
- In chapter 2.4 how were the differences shown? Although Authors refer to some observations, no value is given supporting these observations, only representative images.
Response: To address the reviewer's concern, the quantification graphs of observation including demyelination and nerve derangement have been provided in figure 6 of the revised manuscript. The image J software was used for quantification.
- The same is true for chapters 2.5, and 2.6. For example: "VCR significantly reduced the granular layer of the hippocampal dentate gyrus." How was the significant difference determined? It should be presented as well, similarly to Fig. 13 or 14.
Response: To address the reviewer's concern, the quantification graphs of observation including cellular infiltration, vacuolation, and the inflammatory lesion have been provided in figure 7 of the revised manuscript. The thickness of the dentate gyrus of the hippocampus was measured using image j software as reported previously [1, 2]. The quantification has been provided in figure 8 of the revised manuscript.
- Khan, A., et al., Matrine alleviates neurobehavioral alterations via modulation of JNK-mediated caspase-3 and BDNF/VEGF signaling in a mouse model of burn injury. 2020. 237: p. 2327-2343.
- Khan, A., et al., 7β-(3-Ethyl-cis-crotonoyloxy)-1α-(2-methylbutyryloxy)-3, 14-dehydro-Z Notonipetranone Attenuates Neuropathic Pain by Suppressing Oxidative Stress, Inflammatory and Pro-Apoptotic Protein Expressions. 2021. 26(1): p. 181.